# Research on the Reasonable Grouting Strength of Rock-Like Samples in Different Strengths

**DOI:** 10.3390/ma13143161

**Published:** 2020-07-15

**Authors:** Yan Wang, Zu-qiang Xiong, Chun Wang, Cheng-dong Su, Xue-feng Li

**Affiliations:** 1School of Energy Science and Engineering, Henan Polytechnic University, Jiaozuo 454000, China; 15738513363@163.com (Y.W.); xzqhpu@163.com (Z.-q.X.); scdhpu@163.com (C.-d.S.); 15514076976@163.com (X.-f.L.); 2The Collaborative Innovation Center of Coal Safety Production of Henan, Jiaozuo 454000, China

**Keywords:** shearing experiment, structural plane, reasonable grouting strength, cement mortar samples

## Abstract

One of the key points to make grouting reinforcement technology efficient and economical in engineering is to find out the optimal match relationship between different strengthsofrock mass and strength ofgrouting material. It can be called areasonable grouting strength. To explore that, 5 kinds of *N* (compressive strength of cement mortar) were used to make samples to simulate the different rock strengths. Five kinds of *M* (compressive strength of inorganic double-fluid material) were used to grout the structural plane of samples. Then the shearing experiment was carried out under the same normal stress conditions. Finally, based on the analyses of experimental results, the *C* (the reasonable range of grouting parameters) was proposed. The result shows that when *M* is the same, the smaller *N* is, the higher the *τ*(shearing strength of samples) growth rate is. When *N* is the same, with the increase of *M*, the growth rate of *τ* tends to zero. According to the shearing failure process and morphology of the samples, when *N* is the same, with the increase of *M*, the failure typeofsamples change from plastic flow failure to brittle sliding failure which is the same as the samples without the structural plane. *C* is equal to the ratio of *M* to *N*, and then the evaluation criteria of *C* is established by judging that if it is in the range from 1.19 to *C*_MAX_(*C*_MAX_ = 0.0095*N*^2^ − 0.43*N* + 5.83). It will provide a certain basis to select the *M* in engineering in which *N* is known.

## 1. Introduction

Although some methods are diffusely applied in civil engineering, which provides useful results regarding the seismic design of structures, it is stilleasier to develop some fractures, when the rock mass is disturbed continuously [1,2,3,4,5]. The structural planes of engineering rock mass decrease the bearing capacity of the rock mass. The cracks expand further and connect, which leads to the overall structural failure of the rock mass. To prevent engineering accidents caused by the rock mass with developing structural planes, the grouting reinforcement technique is always widely used to improve the integrity of engineering rock mass and its strength, especially in fields of coal mines and tunnels [6,7,8,9].

As we all know, the mechanical properties of rock mass with structural planes will be changed significantly after grouting, particularly when they are compared with the intact rock mass or rock mass with structural planes without grouting [10,11,12,13,14]. Until now, many scholars had carried out some experiments to analyze the mechanical properties of rock structure after grouting reinforcement. For example, Liu et al. [15] carried out the direct shearing test on the samples with a prefabricated structural plane. The original grouting system was used to ensure the same grouting pressure. The characteristics of shearing shrinkage and shearing expansion of the grouted samples were put forward. Zhang et al. [16] proposed the empirical equation about shearing strength which applied to cement mortar material and the characteristics equation of shearing strength with arctangent function by the direct shearing test. Shen et al. [17] put forward that the shearing strength of the samples was positively correlated with the angle of the structural plane, the expansion characteristics of the structural plane in four different angles, and normal stress. Shi et al. [18] proposed that the shearing strength of the samples increased with the increase of filling thickness by the shearing test.

All the above conclusions are significant for engineering like prevention rib spalling in a coal mine or reinforcement of cracked rock mass in a tunnel. However, few people analyze the relationship between the strength of grouting materials and the strength of grouting bodies. In the engineering of grouting reinforcement for rock mass in different strengths, differentgrouting materialstrengthis selected by the empirical method generally, lacking a theoretical basis. The grouting strength of the material is adjusted mostly by altering the water-cement ratio in the same grouting material with the same external factors such as water and temperature.Therefore, when grouting reinforcement is carried out for different geological bodies, the most matching parameters of them should be found, which can give full play to the maximum effect of grouting materials and greatly save costs.

In this paper, cement mortar in five kinds of compressive strength was used as the material of samples. Inorganic double liquid grouting materials in five kinds of compressive strength were used to grout samples with a prefabricated structural plane. Then the shearing test was carried out under the same normal stress condition. The effects of grouting reinforcement were analyzed by comparing the shearing strength of intact samples, samples with a structural plane, and grouted samples with a structural plane. Finally, the reasonable grouting strength range of different strength rock bodies was proposed for fractured rock masses by discussing the test results further. This study is expected to be used to improve the grouting reinforcement mechanism and give some references in grouting reinforcement engineering.

## 2. Samples Preparation and Experiment Schemes

### 2.1. Design and Preparation of the Structural Plane of Samples

According to the correlational research [19,20,21,22], most scholars in the study of the effect of grouting reinforcement choose cement mortar to make samples, in order to reduce errors during sampling and results of the experiment. The compressive strength of the samples made from cement mortar can ensure that the error is within control. Furthermore, the static load acting on the structural plane can usually be regarded as the resultant force of the shearing load parallel to the structural plane and the normal load perpendicular to the structural plane [21,22,23,24,25]. To eliminate the influence of morphology or angle of structural planes and other variables, the artificial samples were designed with only one horizontal structural plane in the test.

The samples were made from 32.5R cement, sand, and water. After curing for 28 days, the characteristics of the strength of samples with different proportions are shown in Table 1. The temperature in the curing box is about 20 °C and the humidity is higher than 95%.

As is shown in Table 1, the numbers (A~E) are used to represent 5 kinds of water-cement ratios in the experiment. With the proportion of sand and water decreasing, the uniaxial compressive strength of samples increases. The uniaxial compressive strength of 5 kinds of matrix samples is 7 MPa, 10 MPa, 13 MPa, 16 MPa, and 26 MPa, respectively.

The preparation process of samples is shown in Figure 1. The cement mortar prepared by following the proportion was stirred in a mixer and then poured into the molds about 1000 cm^3^ per one. After two hours, the samples were taken out and placed in a standard cement curing box for 28 days. After that, the samples were taken out of the curing box, and the sampleswerecompletely cut along the centerline of the samples with a 3 mm diamond. It means the thickness of the artificial structural plane is 3mm.

### 2.2. Inorganic Double Liquid Grouting Material

The inorganic double liquid grouting material was invented by Professor Xiong at Henan Polytechnic University. The material is composed of two components. Component A is sulphoaluminate cement and additives. The main minerals are C4A3Sand β-C2S. C4A3S can be hydrated in a low hydroxide concentration, forming thick needle AFT (ettringite). The components B is mostly made from gypsum and lime. Before mixing, the single component performance of the material is stable, and there is no segregation and bleeding in a long time (2 h to 6 h). After mixing, the material rapidly produces AFT, and the slurry solidification speed is fast. A large amount of AFT can effectively fill the free space. The early strength of double liquid grouting material is high and solidifies quickly. The early strength increases significantly in a low *W/C* (water-cement ratio).

Five kinds of water-cement ratios of grouting materials were selected in the experiment, and the letters (A, B, C, D, and E) were used to represent five kinds of grouting water-cement ratio (0.6, 0.8, 1.0, 1.2 and 1.5). The compressive strength (curing for 28d) and material consumption of inorganic grouting materials with different W/C ratios are shown in Figure 2.

As shown in Figure 2, according to the strength characteristics of grouting materials, the 28d strength of five water-cement ratio materials is 23.2 MPa, 17.5 MPa, 14.7 MPa, 12.7 MPa, and 9.3 MPa, respectively. The fitting degree of logarithm function between water-cement ratio and material strength is higher. With the increase of the *W/C*, the uniaxial compressive strength of the material decreases.

From the material consumption, the *W/C* of grouting materials increases from 0.6 to 1.5, and the material consumption decreases from 1250 kg/m^3^ to 566 kg/m^3^. Therefore, it is of great significance to choose a reasonable grouting material strength (W/C ratio) for saving the grouting consumption in engineering.

### 2.3. ExperimentalGrouting System

The experimental grouting system was improved based on the cement grouting system created by Liu [25,26]. It becomes suitable for inorganic double liquid grouting material by adding a cut-off valve, tee mixing pipe, and setting mixer in the grouting container.

The improved grouting system can effectively control the water-cement ratio of grouting, ensure the same grouting pressure, reduce manmade errors, and make the grouting conditions in line with actual conditions. The grouting system is shown in Figure 3.

It can be seen from Figure 3 that the pressurization and mixing devices of slurry A are the same as devices of slurry B. The system of slurry A is taken to be an example to explain. Before grouting, the mixer will continue to mix. During grouting, the exhaust valve will be closed, and then the air inlet valve, stop valve, and then pressure valve will be opened to press N_2_ into the grouting pressure vessel. Slurry A and slurry B are mixed in the tee pipe and the double liquid grouting material is used to fill the prefabricated structural plane. After grouting, the pressure valve should be closed and the exhaust valve for pressure reduction should be opened. Two hours later, samples should be taken out and put into the curing box for 28 days. After that, samples should be taken out and polished.

According to the above preparation method, there are three kinds of samples required for the experiment, i.e., intact samples, samples without grouting on the prefabricated structural plane, and grouted samples with the prefabricated structural plane. Three samples are shown in Figure 4.

### 2.4. Shearing Experimental System and Scheme

RMT-150B electro-hydraulic servo rock mechanics experimental system (developed by Wuhan Institute of geotechnical engineering, Chinese Academy of Sciences) was adopted in this experiment.

The shearing unit system of the experimental system is shown in Figure 5, which is mainly composed of a hydraulic cylinder, the loading column, the push-pull device, the displacement sensor, the shearing box, and the support plate. The roller plate is added to the above samples to eliminate the influence of friction between the rock sample and the device during the shearing process.The shearing experiment scheme is shown in Table 2.

In the experiment, three kinds of samples were used to carry out the shearing experiment respectively. The vertical load control method was adopted in the test. The normal load rate was set as 0.5~1 kN/s, and the normal load was set as 25% of the compressive strength of the sample. Then the horizontal load was set as a displacement control method, the displacement load rate was 0.02 m/s, and the maximum shearing displacement was set as 5 mm.

## 3. Results of the Shearing Experiment

### 3.1. Controlled Experimental Results

To analyze the effect of the grouting, the shearing experiment was carried out for the intact samples and the samples without grouting under the same conditions and normal stress. The results are shown in Figure 6.

As shown in Figure 6, under the same normal stress condition, the shearing strength of intact samples increases with *N* (the strength of the matrix sample), from 3.26 MPa to 14.66 MPa, and the cohesion increases from 2.13 MPa to 4.83 MPa. The internal friction angle increases from 33.55° to 56.13°.

The shearing strength of the sample without grouting on the prefabricated structural plane increased from 0.66 MPa to 5.07 MPa. Both the cohesion and internal friction angle of the samples show a slightly higher trend under the influence of the increase of normal stress.

According to the characteristics of the above data, we can obtain the following:(1)With the increase of *N*, cohesion, and internal friction angle of the intact samples increase. Because samples without grouting are mainly affected by the normal stress, though the strength of samples increased, the internal friction angle and cohesion are only slightly increased(2)Under the same condition of normal stress, the shearing strength of the sample only accounts for 18.33% to 34.58% of the strength of intact samples, and the cohesion only accounts for 1.6% to 2.9% of cohesion of the intact samples.

### 3.2. Results of the Shearing Experiment of Grouting Samples on the Structural Plane

The shearing experiment was carried out, after two liquid grouting materials in 5 kinds of water-cement ratio (0.6, 0.8, 1.0, 1.2, 1.5) were used to grout the prefabricated structural plane of samples in 5 kinds of compressivestrengths of samples (7 MPa, 10 MPa, 13 MPa, 16 MPa, 26 MPa). According to the experiment results, the scatter diagram of shearing strength results is drawn, as shown in Figure 7.

The shearing strength of the grouted samples is in the range of 2.27 MPa to 8.73 MPa. The shearing strength of the samples increases continuously when the strength of the grouting material and *N* increase.

To analyze the influence of water-cement ratio and strength of samples on the shearing strength of the samples, the data are projected on the X-Z and Y-Z planes of the three-dimensional coordinate axis, respectively.The numbers above the colored points mean shearing strength of samples.

The relationship between the water-cement ratio of grouting material and the shearing strength of the samples is shown in Figure 8 and the relationship between the matrix strength of samples and the shearing strength of the samples is presented in Figure 9.

As shown in Figure 8, marks of the same color represent samples with the same *N*, from left to right are 7 MPa, 10 MPa, 13 MPa, 16 MPa, and 26 MPa; marks from top to bottom represent 5 kinds ofW/C (water-cement ratio) of grouting materials, i.e., 0.6, 0.8, 1.0, 1.2 and 1.5. The shearing strength characteristics of *W/C* from Figure 8 are as follows:(1)Under the same normal stress, with the decrease of *W/C* (the strength of grouting material increases), the *τ*(the shearing strength of the samples) increases.(2)The larger the *N* is, the greater the influence of the increasing *W/C* is on the sampleswith the grouted structural plane.(3)By fitting the data of the samples in the same *W/C*, the curves are all following the characteristics of an exponential function. That means with the increase of *N*, the *τ*increases continuously, and the reduction rate is close tozero gradually.

From Figure 9, the characteristics of *N* and *τ* are as follows:(1)For the samples in the same *N*, with the increase of *W/C*, the *τ* decreases continuously. The reduction rate of *τ* increases, which is more significant for the samples with larger *N*.(2)The smaller the *W/C* is, the greater the difference of *τ* is in different *N*.(3)The fitting degree of the quadratic function is high when *N* and *τ* are fitted. With the decrease of W/C (the strength of grouting material increases), *τ* increases slowly and finally tends to be steady.

According to the analysis of the characteristics in Figure 8 and Figure 9, the upper limit of *τ* is affected by the *N* and *W/C*. When one factor is changed unilaterally (*W/C* reduces or *N* increases), the growth rate of *τ* decreases gradually and tends to zero. 

## 4. Analysis of Grouting Reinforcement Effect and Failure Characteristics of Samples

Comparing the shearing strength of the grouted samples can analyze the grouting effect. Analyzing the failure process of samples and the shape of shearing failure of the samples can provide the basis for analyzing the reasonable parameters of grouting for different strength rock masses [27,28,29,30]. Therefore, further analysis is carried out from the following two aspects.

### 4.1. Grouting Effect of the Samples with the Prefabricated Structural Plane

Three kinds of *N* are chosen, i.e., 7 MPa, 16 MPa, and 26 MPa, respectively. 

In Figure 10a, *τ* of samples increases significantly when the *N* is 7 MPa. The shearing strength of the grouted samples in five kinds of *W/C* increases significantly with an increase of about 243.18% to 343.18% by comparing the strength of samples without grouting, and the peak strength is close to the *τ* of the strength of intact samples. 

With the increase of *W/C* from 0.6 to 1.5, the shearing strength of the samples decreased from 2.97 MPa to 2.27 MPa.

As Figure 10b shows, with the decrease of *W/C*from 1.5 to 0.6, the *τ* of the grouted samples in which the *N* is 16 MPa increases from 3.59 MPa to 6.30 MPa. The *τ* of grouted samplesis about half of *τ* of intact samples, with an increase of about 13.97% to 100%. 

Figure 10c shows that, when the *W/C* of samples in which the *N* is 26 MPa decreasing from 1.5 to 0.6, the *τ* increases from 2.38 MPa to 4.02 MPa. The *τ* is less than half of the shearing strength ofintact samples, with an increase from 8.78% to 72.19%. 

Overall, the lower the *N* is, the closer the *ττ* of grouted samples are to the strength of the intact samples. When *W*/*C* is the same, *N* increases, the *τ* increased slowly and the grouting effect decreases.

### 4.2. Characteristics of Shearing Strength of Grouted Sample on Prefabricated Structural Planes

The characteristics were summarized by analyzing the influence of different *N* and different strengths of grouting material (*M*) on the shearing strength of the grouted samples under the same stress condition.

Figure 11 shows the fitting of the characteristic curve of *M* and *τ*, using the normalized value (*U*) to express the ratio of the shearing strength of the grouted samples to the shearing strength of the intact samples.

As shown in Figure 11a, the quadratic function is fitted which represented the relationship between *τ* and *M* when the *N* is 7 MPa.
(1)$$\tau_{7}=-0.0033M^{2}+0.16M+1.08$$
(2)$$\tau_{7}\prime =-0.0066M+0.16$$

Equation (1) is the quadratic fitting function of *τ*_7_. The correlation coefficient that the *R*^2^_7_ = 0.9706, meaning the high fitting degree of the function.

Equation (2) is a linear function which is the first derivative of Equation (1). It indicates the growing trend of *τ*_7_. The larger *τ*_7_′ is, the faster *τ*_7_ growth is. The coefficient of the primary term of Equation (2) is −0.0066, less than 1.
(3)$$\tau_{max}=\frac{-B+\sqrt{B^{2}-4AC}}{2A}$$

In Equation (3), *A* represents the coefficient of the quadratic term in Equation (1), B represents the coefficient of the primary term, and C represents the constant. It can be seen from Equation (1) parameter that when *M* is 23.86 MPa, *τ*_7_ is expected to reach the maximum.

The test results show that *M* increases from 9.3 MPa to 23.2 MPa and *U* increases from 0.6 to 0.9. It shows that 23.2 MPa is close to the prediction result of 23.86 MPa, so the *U* tends to 0.9, which shows that the grouting strength of the 7 MPa mortar block samples reaches 23.86 MPa, the *τ* tends to remain unchanged, only reaching 90% of the strength of the intact samples.The principle of subsequent analysis is consistent with that here.

As shown in Figure 11b, the quadratic function was fitted which represented the relationship between *τ* and *M* when the *N* was 16 MPa.
(4)$$\tau_{16}=-0.012M^{2}+0.60M-0.89$$
(5)$$\tau_{16}\prime =-0.024M+0.60$$

Equation (3) is the quadratic fitting function of *τ*_16_. The fitting degree of the function is high due to the correlation coefficient (R^2^_16_ = 0.9637).

Equation (5) is a linear function which is the first derivative of Equation (4). It indicates the growing trend of *τ*_16_.The larger *τ*_16_′ is, the faster *τ*_16_ growth is. The coefficient of the primary term of Equation (5) is −0.0024, less than 1.

From Equation (3) by taking the parameters in Equation (4) into Equation (3), the *τ*_16_ reached the maximum value when *M* was 24.22 MPa. The experiment results show that, when the *M* is 23.2 MPa, the maximum *U* value is 0.81. Because 23.2 MPa is close to the predicted maximum value (22.8 MPa), the predicted maximum *U* value tends to 0.81.

As shown in Figure 11c, the quadratic function is fitted which represented the relationship between *τ* and *M* when the *N* is 16 MPa.
(6)$$\tau_{26}=-0.015M^{2}+0.72M-0.015$$
(7)$$\tau_{26}\prime =-0.030M+0.72$$

Equation (6) is the quadratic fitting function of *τ*_26_. The fitting degree of the function is high due to the correlation coefficient (R^2^_16_ = 0.9799).

Equation (7) is a linear function which is the first derivative of Equation (6). It indicates that the larger *τ*_26_′ is, the faster *τ*_26_ growth is. The coefficient of the primary term of Equation (7) is −0.03, less than 1.

From the calculation of Equation (3) by taking the parameters in Equation (6) into Equation (3), *τ*_26_ reached the maximum value of 8.65 MPa when *M* is 24.41 MPa. The experiment results show that when the *M* is 23.2 MPa, the maximum *U* value is 0.81. Because 23.2 MPa is close to the predicted maximum value (24.41 MPa), the predicted maximum *U* value tends to 0.81.

In summary, the following characteristics can be obtained:(1)Grouting can effectively improve the shearing strength of samples with the structural plane, but no matter how big the *M* is, it cannot reach the strength value of the complete sample.(2)The relationship between *τ* and *M* is quadratic. With the increase of *M*, the growth rate of *τ* decreases gradually.(3)With the increase of *N*, the upside of the effect of grouting reinforcement is becoming limited. The *U* decreases from 0.9 to 0.6.

### 4.3. Shearing Failure Process and Morphology Characteristics of Grouted Samples

The failure process and the failure morphology characteristics of grouted samples can reflect the effect of grouting reinforcement, which provides a certain basis for selecting the reasonable parameters of grouting. Therefore, the morphology of failure and the shearing strength-displacement curve of samples with different *M* were analyzed. The samples in which the *N* was equal to 16 MPa were taken as examples to analyze, as shown in Figure 12.

InFigure 12, at the initial stage of load growth, the shearing stress growth rate of the samples without grouting is the biggest, and the shearing displacement curve is approximate to a straight line. With the increase of *M*, the growth rate of shearing stress gradually decreases from 12.7 MPa to 17.5 MPa. However, all their strengths are smaller than that of samples without grouting, and they have shown a trend of fluctuation growth. The growth rate of the shearing stress of the intact samples is between samples without grouting and grouted samples. It shows a downward convex trend at the initial stage of shearing.

Figure 11 shows thatthe peak strength is not obvious in the sample without grouting on the prefabricated structural plane, and the sample grouted in which the *M* was 12.7 MPa. Although the shearing strength of the sample is improved after the grouting, the failure types are still the same, both of which are a plastic failure. When *M* increases to 14.7 MPa, the strength of the sample begins to show anunobvious peak value, which tends to alter from plastic failure to brittle failure. When *M* increased to 17.5 MPa, there was an obvious peak strength and showed brittle failure. 

The intact samples show the brittleness. The peak strength of the sample is obvious, and then the strength drops sharply. The residual strength of the sample is larger than the sample without grouting on the structural plane and smaller than that of the grouted samples. 

The shearing failure morphology of samples in different *M* is shown in Figure 13.

Figure 13 shows the samples in which the *N* was 16 MPa. From left to right, i.e., D1, Da, Db, Dc. The failure morphology of these four samples is almost the same. As there is no obvious damage trace, they are replaced by one appearance.

From Figure 13, the surface of the rock sample (D1) is intact with obvious friction traces on the surface. When *M* is in the range from 9.3 MPa to 14.7 MPa, the failure mainly occurred at the level of grouting material, which is characterized by plastic failure. When *M* is 17.5 MPa, the shearing failure occurred on the side of the sample. The bonding surface between the grouting layer and the mortar block is damaged with the tensile failure feature locally. The failure type begins to change from plastic sliding to brittle shearing sliding.

In summary, in the samples after grouting, not only was their shearing strengths improved, but also the failure characteristics of the samples with the structural plane were changed, which made it change from plastic failure to brittle shearing failure. The failure type was gradually close to the intact samples and had residual strength among them.

## 5. Discussion on the Reasonable Grouting Strength of Fractured Rock Mass

Although grouting can improve the strength of rock mass with structural planes, the strength of complete rock mass cannot be reached. The effect of grouting will not improve when *M* exceeds a certain range. Therefore, in engineering applications, it is of great significance to select reasonable grouting strength for different rock masses. If the *M* is too large, it will cause a waste of materials; if the *M* is too small, it may not be able to effectively strengthen the rock mass.

*C* is set as a reasonable grouting strength in the experiment, which satisfies the following relationship.
(8)$$C=\frac{M}{N}$$

In Formula (8), *M* is the strength of grouting material, MPa; *N* is the strength of the samples, MPa. Among the five groups of samples, the smallest *M* in which the samples show brittle failure is selected as the lower limit of *C*. The *M* which made the *τ* biggest in fitting function is selected as the upper limit of *C*. According to the above standards, *C* of 5 kinds of grouted samples is identified.

In the scope of this experiment, the characteristics of *C* of the samples in five kinds of strength are shown in Figure 14.

As shown in Figure 14, within a reasonable range, the failure type of the samples after grouting is the same as that of the complete sample. Beyond this range, the effect of grouting reinforcement tends to be stable. Even if the *M* continues to increase, the growth rate of *τ*is very slow.

When *C* < 1.19, the grouting reinforcement effect of the samples increases obviously, which shows that when *N* is the same *M* increases which will cause *τ*to increase rapidly. 

It is the reasonable choice for *C* when 1.19 ≤ *C* ≤ *C_MAX_*. The number 1.19 is the lower limit of *C*; at this time, the failure characteristics are consistent with that of the intact samples for the first time. The number *C_MAX_* is the upper limit of *C*, which satisfies the feature of the quadratic function. With the increase of *N*, the *C*_MAX_ increases.

When *C* is bigger than *C*_MAX_, the grouting reinforcement effect of the sample tends to be stable, *M* continues to increase, and the growth rate of *τ* approaches zero.

## 6. Conclusions

(1)*N* is in the range from 7 MPa to 26 MPa, and *M* is in the range from 9.3 MPa to 23.2 MPa. No matter which combination of *M* and *N*, the shearing strength of samples cannot reach the shearing strength of the intact samples.(2)The increase of *M* can enhance the effect of grouting reinforcement. The lower *N* is, the more obvious the reinforcement effect is. With the increase of *N*, the influence of the increase of *M* on the strength of samples decreases. With the increase of *N*, the growth rate of *τ* decreases.(3)When *M* increases, the failure morphology of the grouted samples is consistent with the plastic failure of the sample without grouting at the initial stage, and it gradually changes to the brittle shearing slip failure which is like that of intact samples.(4)The *C* is reasonable in the range (1.19 ≤ *C* ≤ *C*_max_). When *C* is less than 1.19, *τ*increases obviously with *M* increasing. When *C* is larger than *C_MAX_*, the grouting reinforcement effect will not be improved, and if *M* increases continuously, the cost of engineering will increase. Thus, *C* can provide a reference for grouting reinforcement in engineering.

## Figures and Tables

**Figure 1 materials-13-03161-f001:**
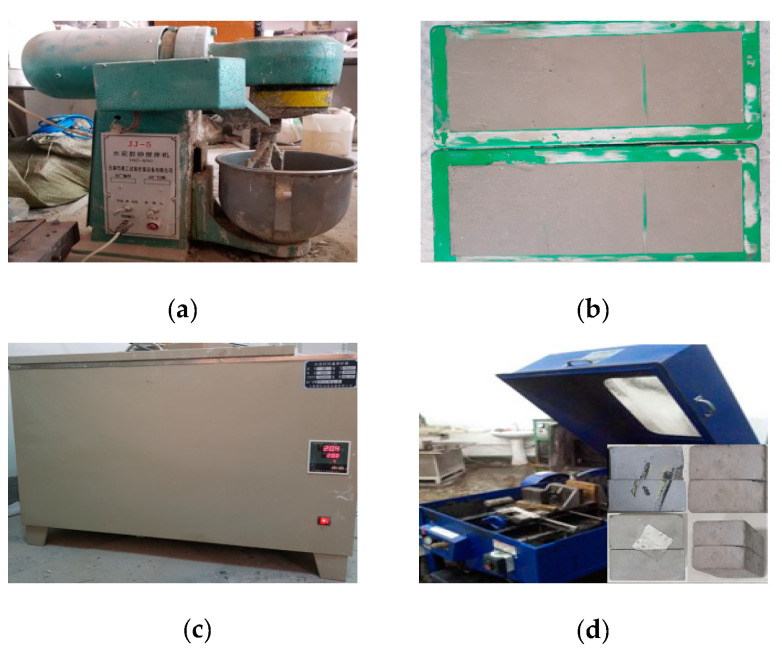
The preparation process of samples. (**a**) Cement mortar was stirred in a mixer. (**b**) Cement mortar was poured into molds. (**c**) Samples in a standard cement curing box. (**d**) Samples were cut.

**Figure 2 materials-13-03161-f002:**
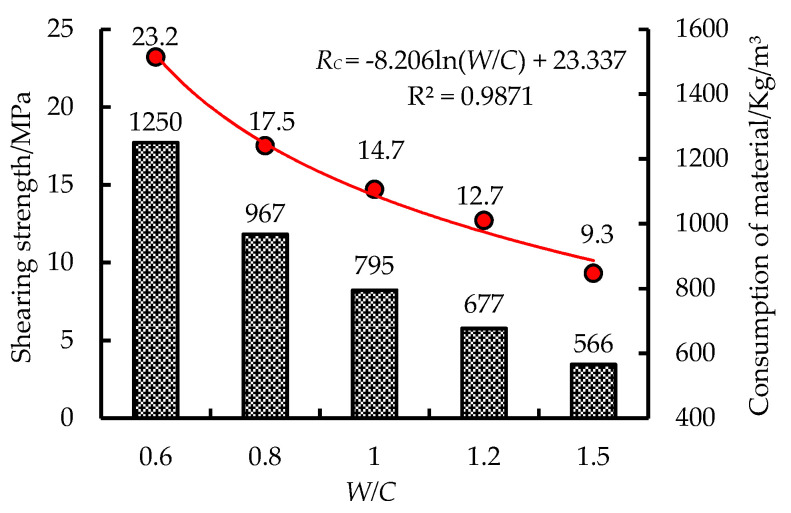
Strength and consumption of grouting materials with different water-cement ratios.

**Figure 3 materials-13-03161-f003:**
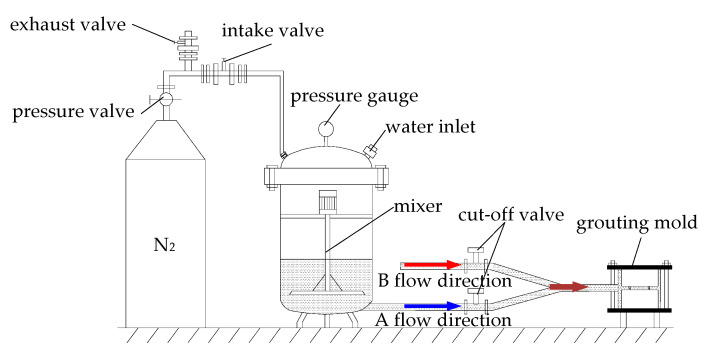
Grouting system of pressure-retaining for structural face rock specimen.

**Figure 4 materials-13-03161-f004:**
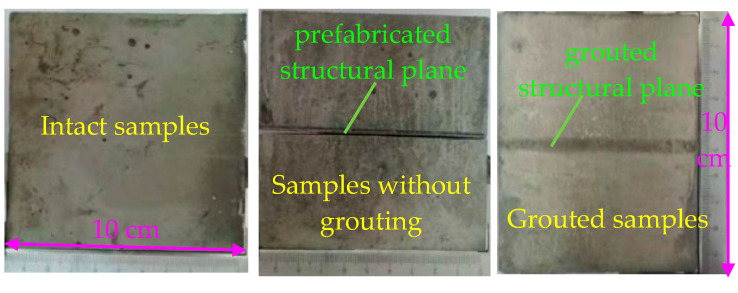
Prefabricated mortar block with structural surface.

**Figure 5 materials-13-03161-f005:**
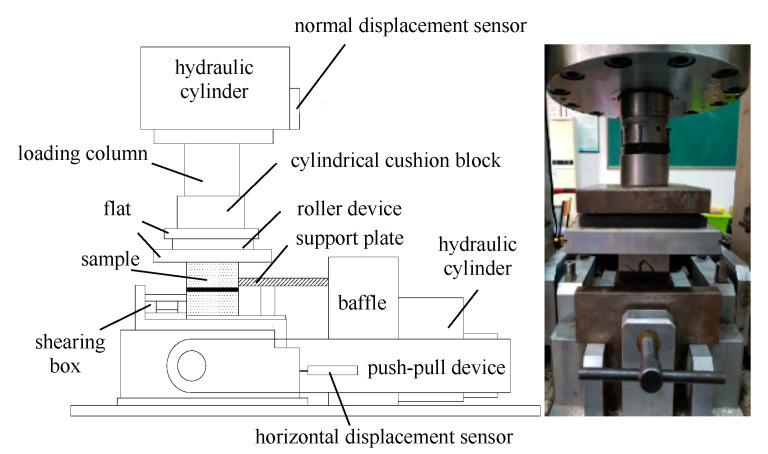
The test system of shearing loading of RMT-150B.

**Figure 6 materials-13-03161-f006:**
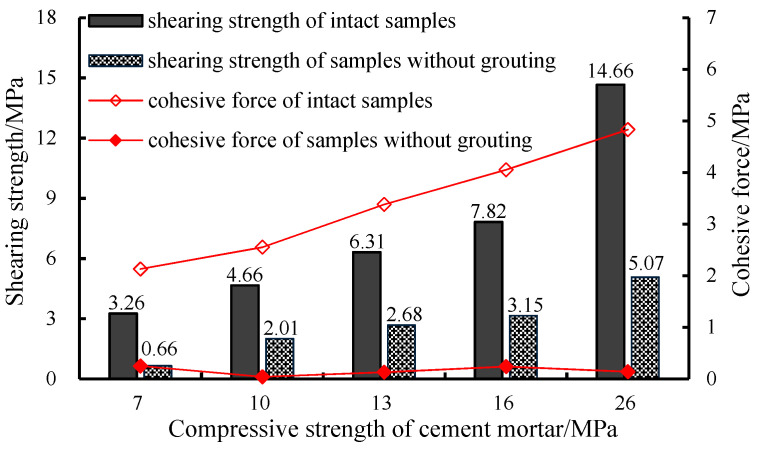
Results of the shearing experiment in the controlled group.

**Figure 7 materials-13-03161-f007:**
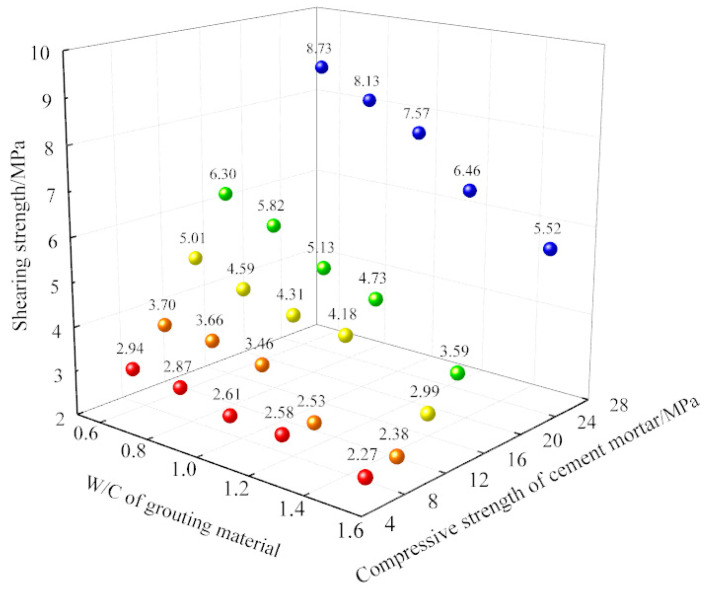
Results of the experiment of samples with a grouted structural plane.

**Figure 8 materials-13-03161-f008:**
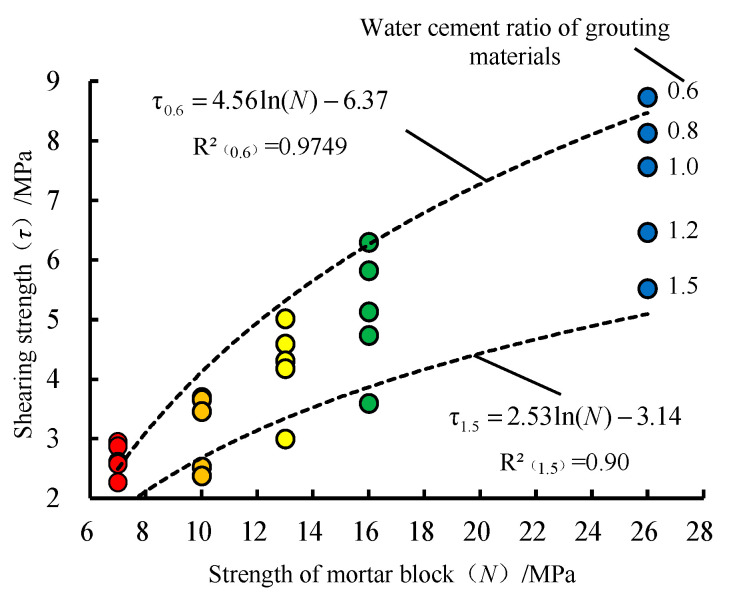
Characteristics between water-cement ratio(*W/C*) and *ττ*.

**Figure 9 materials-13-03161-f009:**
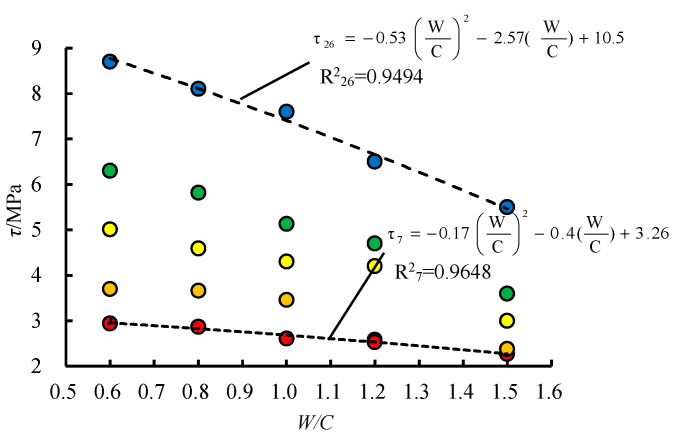
Characteristics between *N* and *ττ*.

**Figure 10 materials-13-03161-f010:**
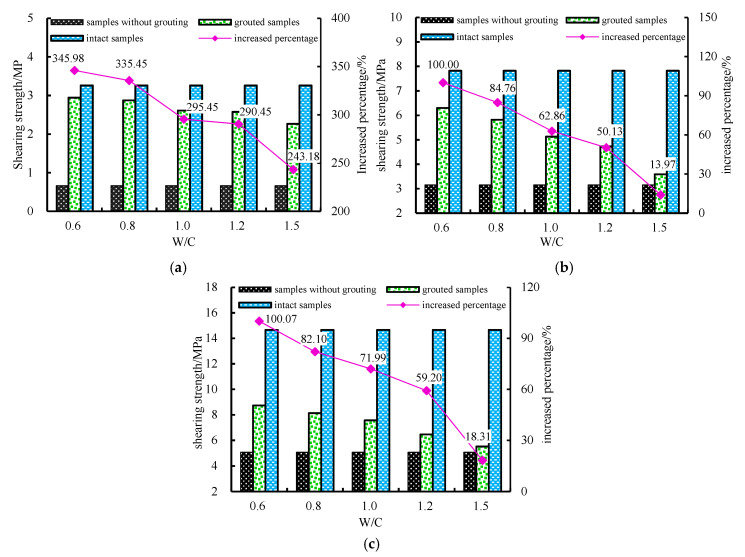
Grouting effect of *N* in different *W/C*: (**a**) 7 MPa samples grouting effect; (**b**) 16 MPa samples grouting effect; (**c**) 26 MPa samples grouting effect.

**Figure 11 materials-13-03161-f011:**
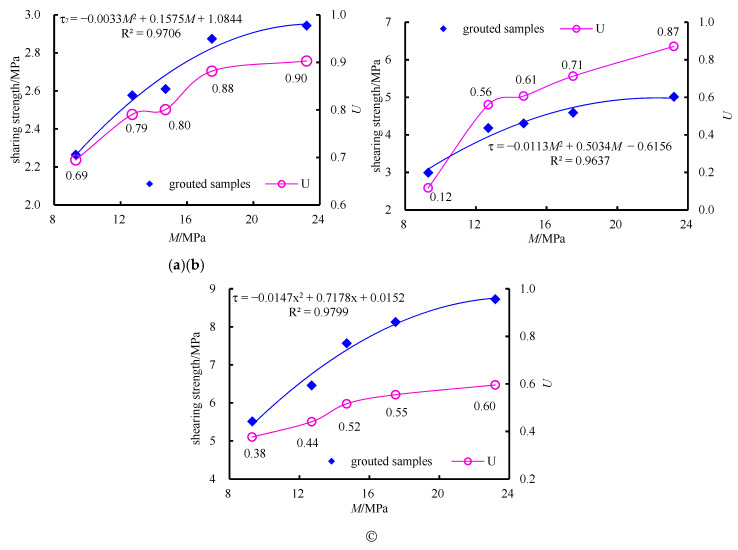
Growth characteristics of shearing strength of samples in different *M*: (**a**) Growth characteristics of shearing strength of 7 MPa samples; (**b**) growth characteristics of shearing strength of 16 MPa samples; (**c**) growth characteristics of shearing strength of 26 MPa samples.

**Figure 12 materials-13-03161-f012:**
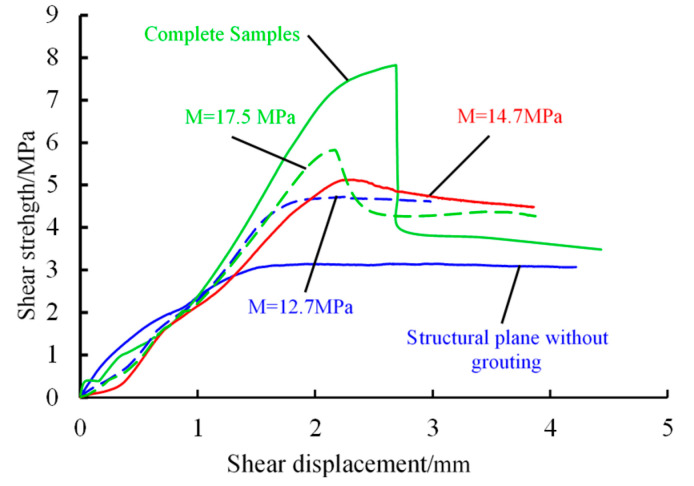
The curve of shearing strength and shearing displacement.

**Figure 13 materials-13-03161-f013:**
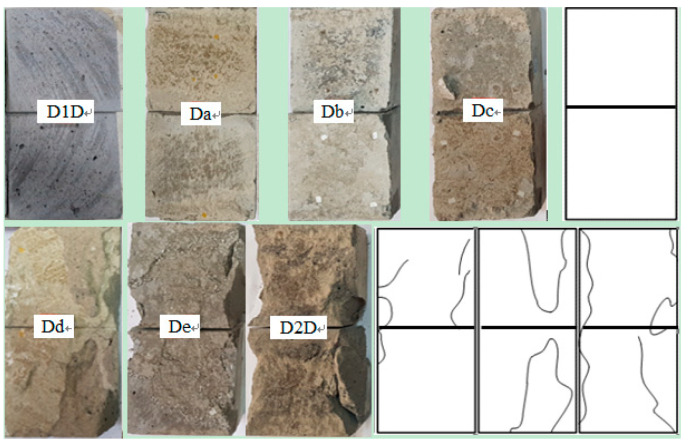
Failure characteristics of the samples.

**Figure 14 materials-13-03161-f014:**
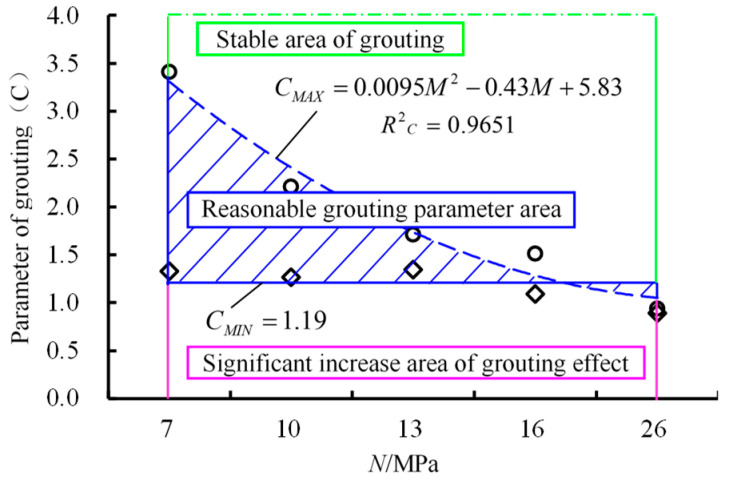
Characteristics of reasonable grouting strength.

**Table 1 materials-13-03161-t001:** Uniaxial compressive strength and cement mortar ratio of matrix material.

Number	Proportion of Sand	Proportion of Cement	Proportion of Water	Uniaxial Compressive Strength/MPa
A	4.5	1	1	7
B	4	1	0.9	10
C	3.5	1	0.8	13
D	3	1	0.7	16
E	2	1	0.6	26

**Table 2 materials-13-03161-t002:** The shearing experiment scheme.

Group	Number	Strength of Samples/MPa	Normal Load/MPa	Strength of Grouting Material/MPa
Intact samples	A1	7	1.75	/
	B1	10	2.5	/
	C1	13	3.25	/
	D1	16	4	/
	E1	26	6.5	/
Samples without grouting	A2	7	1.75	/
	B2	10	2.5	/
	C2	13	3.25	/
	D2	16	4	/
	E2	26	6.5	/
Grouted samples	Aa	7	1.75	9.3
	Ab			12.7
	Ac			14.7
	Ad			17.5
	Ae			23.2
	Ba	10	2.5	9.3
	Bb			12.7
	Bc			14.7
	Bd			17.5
	Be			23.2
	Ca	13	3.25	9.3
	Cb			12.7
	Cc			14.7
	Cd			17.5
	Ce			23.2
	Da	16	4	9.3
	Db			12.7
	Dc			14.7
	Dd			17.5
	De			23.2
	Ea	26	6.5	9.3
	Eb			12.7
	Ec			14.7
	Ed			17.5
	Ee			23.2

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
