# Peer review of "Research on the Reasonable Grouting Strength of Rock-Like Samples in Different Strengths"

_materials, 2020, doi:10.3390/ma13143161_

Round 1

Reviewer 1 Report

In this study inorganic double liquid grouting materials in five kinds of strength were used to grout the samples with prefabricated structural planes. Then the shearing test was carried out under the same normal stress condition.

The topic is very intersting and several experimental results are presented by the authors. The paper is also well written and organized. 

In order to improve the manuscript, the bibliography and the introduction should be revised by adding more references related to the same topic and future research directions could be included in the conclusions section.  Some examples are:

A discrete-to-continuum approach to the curvatures of membrane networks and parametric surfaces. Mechanics Research Communications56, 18-25, 2014.

Loading noise effects on the system identification of composite structures by dynamic tests with vibrodyne. Composites Part B: Engineering115, 376-383, 2017.

Author Response

The statement revised to the opinion of the reviewer 1

Thank you for your letter and for the reviewers’ comments concerning our manuscript entitled “Research on the reasonable grouting strength of rock-like samples in different strength”. Those comments are all valuable and very helpful for revising and improving our paper, as well as the important guiding significance to our researches. We have studied comments carefully and have made a correction which we hope meet with approval. Revised portions are marked in blue in the paper. The main corrections in the paper and the response to the reviewer’s comments are as flowing:

Thank you for your careful review, the author has modified the manuscript based on your comments.

  1. In order to improve the manuscript, the bibliography and the introduction should be revised by adding more references related to the same topic,and future research directions could be included in the conclusions section.

Response:Accordingtorecommendations ofreviewer,references related to the topic hasbeendownloadandread. The research methods and conclusions bring some wise ideas. Thus the author has revised the introduction, and add the references in this manuscript.

Reviewer 2 Report

The article is interesting but requires minor corrections.

My remarks:

  1. You need to standardize the notation (Figure or Fig.), for example Figure 7, 8, 13
  2. line 185, 3.1 Results of the shearing experiment of grouting samples on the structural plane. Incorrect numbering of the point.
  3. line 316, 4.2 Shearing failure process and morphology characteristics of grouted samples. Incorrect numbering of the point.
  4. Figure 12 and Figure 14. Illegible description of the ordinate axes.

Author Response

The statement revised to the opinion of the reviewer 2

Thank you for your letter and for the reviewers’ comments concerning our manuscript entitled “Research on the reasonable grouting strength of rock-like samples in different strength”. Those comments are all valuable and very helpful for revising and improving our paper, as well as the important guiding significance to our researches. We have studied comments carefully and have made a correction which we hope meet with approval. Revised portions are marked in blue in the paper. The main corrections in the paper and the response to the reviewer’s comments are as flowing:

Thank you for your careful review, the author has modified the manuscript based on your comments.

1.You need to standardize the notation (Figure or Fig.), for example, Figure 7, 8, 13

Response:Thanks to the reviewer for carefully reviewing,he notation has standardized in Fig. , for example, Fig. 7。

  1. line 185, 3.1 Results of the shearing experiment of grouting samples on the structural plane. The incorrect numbering of the point.

Response:The numbering of point 3.1 inline 185 hasbeencorrectedinto 3.2

  1. line 316, 4.2 Shearing failure process and morphology characteristics of grouted samples. The incorrect numbering of the point.

Response:The numbering of point 4.2 inline 316 has been corrected into4.3

  1. Figure 12 and Figure 14. Illegible description of the ordinate axes.

Response:The descriptions of the ordinate axes are illegible because of the wrong format in Figure 12 and Figure 13. They have been modified.

Reviewer 3 Report

Already at the beginning of the article, there are ambiguities that may mislead the reader. The Authors wrote:

“To explore that, 5 kinds of N (strength of cement mortar) were used to make samples to simulate the different strength rock. And 5 kinds of M (strength of inorganic double-fluid material) were used to grout the structural plane of samples. Then the shearing experiment was carried out under the same normal stress conditions. Finally, based on the analysis of experiment results, The C (reasonable range of grouting parameters) is proposed.”

- what exactly does “strength of cement mortar” mean? It should rather be: “compressive strength of cement mortar”, I think. There is more ambiguity later in the manuscript.

The Authors wrote:

„The correlational researches[17-20] show that samples made from cement mortar can avoid the errors caused by various and irregular cracks of the rock mass.”

- I have not found such a statement in the cited literature

After that, the samples were taken out of the curing box, and the prefabricated structural plane was cut along the centerline of the samples with a 3 mm diamond. – to what depth the cuts were made??

In Figure 6 it is difficult to distinguish: “Complete samples” and “Samples without grouting”.

As shown in Figure 6, under the same normal stress condition, the shearing strength of the complete samples increases with N(the strength of the matrix sample), from 3.26MPa to 14.66MPa, and the cohesion increases from 2.13MPa to 4.83MPa.

 - In Figure 6, the value of "increasing cohesion" is not shown. It is also unknown what the lines, described as „Complete samples” and “Samples without grouting”, represent.

What does "Strength of samples/MPa" mean? What do the numbers above the colored points mean? Please explain in detail what the drawing shows.

I suggest you read it carefully and improve all the work.

Author Response

The statement revised to the opinion of the reviewer 3

Thank you for your letter and for the reviewers’ comments concerning our manuscript entitled “Research on the reasonable grouting strength of rock-like samples in different strength”. Those comments are all valuable and very helpful for revising and improving our paper, as well as the important guiding significance to our researches. We have studied comments carefully and have made a correction which we hope meet with approval. Revised portions are marked in blue in the paper. The main corrections in the paper and the response to the reviewer’s comments are as flowing:

Thank you for your careful review, the author has modified the manuscipt based on your comments.

  1. what exactly does “strength of cement mortar” mean? It should rather be: “compressive strength of cement mortar”, I think. There is more ambiguity later in the manuscript.

Response:Because of the negligence of the author, the description of “strength of cement mortar” is not clear. All of them in the manuscript have corrected into “compressive strength of cement mortar”.

  1. The correlational researches[17-20] show that samples made from cement mortar can avoid the errors caused by various and irregular cracks of the rock mass.”, which have not been found such a statement in the cited literature.

Response:Depending on the question of the reviewer, the author finds that there is ambiguity in writing this sentence. It should be expressed that, “according to the relevant researches, most scholars in the study of the effect of grouting reinforcement, choose cement mortar to make samples, in order to reduce the errors during sampling and results of the experiment. The compressive strength of the samples made from cement mortar can ensure that the error is within control”. The above has been revised in this manuscript and the references have been revised.

  1. After that, the samples were taken out of the curing box, and the prefabricated structural plane was cut along the centerline of the samples with a 3 mm diamond. – to what depth the cuts were made?

Response:The cement mortar sample is completely cut along the middle line usinga 3mm diamond, and the thickness of the structural plane is 3 mm. It has been modified in the manuscript.

  1. In Figure 6, the value of "increasing cohesion" is not shown. It is also unknown what the lines, described as „Complete samples” and “Samples without grouting”, represent.?

Response:Depending on the question of the reviewer, I found a problem with my drawing, so I redrawn Figure 6, modified the description of the horizontal coordinates, and perfected the legend. The figure is mainly to show shearing test results of the intact samples of cement mortar and samples having a structural plane without grouting. By analyzing the results of this set of tests, it can be used for the analysis of the grouting reinforcement effect.

  1. What does "Strength of samples/MPa" mean? What do the numbers above the colored points mean? Please explain in detail what the drawing shows.

Response:The axis description of Figure 7 has been modified,"Strength of samples/MPa" means “compressive strength of cement mortar” Compressive strength of cement mortar is equal to the compressive strength of intact samples. The numbers above the colored points mean shearing strength of samples. The description of them has been added to the manuscript.

Round 2

Reviewer 3 Report

No more comments